# Research on the Energy Release Characteristics of Six Kinds of Reactive Materials

**DOI:** 10.3390/ma12233940

**Published:** 2019-11-28

**Authors:** Xianwen Ran, Liangliang Ding, Jingyuan Zhou, Wenhui Tang

**Affiliations:** College of Liberal Arts and Sciences, National University of Defense Technology, Changsha 410073, China; zhoujingyuan19@nudt.edu.cn (J.Z.); tangwenhui@nudt.edu.cn (W.T.)

**Keywords:** reactive material, PTFE, energy release characteristics, drop hammer, quasi-closed container

## Abstract

Currently, PTFE/Al is widely used in the reactive fragmentation warhead. However, for the same explosive yield, the reactive fragments usually have a smaller damage-radius than the inert fragments because PTFE/Al has a poor penetration ability and needs an impact-speed up to 1000 m/s to stimulate its chemical reaction. To enhance the damage power of reactive fragments, six kinds of reactive materials (PTFE/Al, PTFE/B, PTFE/Si, PTFE/Al/B, PTFE/Al/Si, and PTFE/Al/CuO) based on PTFE were designed and studied. Through the drop weight system and the self-designed energy release test device, qualitative and quantitative analysis of the energy release ability of six kinds of reactive materials were carried out. The qualitative analysis results indicate that the reactions of PTFE/B and PTFE/Si are weak under the impact of drop hammer with only a very weak fire light produced, while the reactions of PTFE/Al, PTFE/Al/B, PTFE/Al/Si, and PTFE/Al/CuO are relatively intense, and the reaction of PTFE/Al/Si is the most intense. Through the self-designed energy release test device, the energy release ability of the reactive material was quantitatively compared and analyzed. The results show that the energy release ability of the four formulations were as follows: PTFE/Al/Si > PTFE/Al/CuO > PTFE/Al/B > PTFE/Al. Therefore, it can be concluded that the PTFE/Al/Si formulation is a new reactive material with strong energy release ability, which can be a new choice for reactive fragment.

## 1. Introduction

Reactive material has been widely studied for more than 40 years [1] and the metal/polymer mixture type reactive materials showed great superiority in the application of weapons and equipment [2]. Metal/polymer mixture type reactive materials are usually composed of two or more metal particles and polytetrafluoroethylene (PTFE). This type of materials is relatively insensitive under normal conditions, but, under high strain rate loading or high-pressure collision conditions, violent chemical reactions will occur and release a lot of chemical energy [3,4]. In 2004, the scientists in the “naval weapon operation center” of the US army found that the damage area on the air target resulted from the reactive fragment was nearly doubled if the reactive and inert fragments have the same size and collision velocity. Some characteristics of the metal/polymer mixture type reactive material are similar to conventional explosives, but the reactive material has better mechanical properties than the explosive, and the overall performance of the external impact loading conditions is blunt, high energy. 

Currently, the most widely used PTFE based reactive materials composite formulation is PTFE/Al (73.5%/26.5%), and its unit mass energy and unit volume energy can reach 3.5 times and 5 times tri-nitro-toluene (TNT) explosives [5]. Under the condition of high-speed impact loading, PTFE/Al will undergo violent combustion and detonation reaction, and the reaction rate is roughly between the combustion rate of propellant and the detonation rate of explosive [6,7]. The heat released by the unit mass reaction of PTFE/Al (wt.%: 73.5/26.5) is about 8.53 kJ/g, which is about two times the TNT explosive under the same conditions. To extend the applied scope of PTFE/Al, many works on fabrication method and formulation have been carried out. The tensile strength of PTFE/Al can be increased to 20 MPa after sintering, and the sintered PTFE/Al reactive material can be used as a structural member. The density of PTFE/Al can be increased to more than 10 g/cm^3^ by adding more heavy-metal powder such as tungsten, and then the penetration ability can be enhanced. Cai [8,9] carried out many mechanical property tests on PTFE/Al and PTFE/Al/W, including quasi-static compression test, drop impact test, and dynamic SHPB test. The results show that, when the particle size of Al powder is 2 μm, a small part of PTFE/Al sample will react in the process of quasi-static compression. As for PTFE/Al/W reactive material, when the particle size and porosity between particles decrease, the compressive strength of the formed sample will increase. In addition, the mechanical behavior of PTFE/Al/W reactive material under the constraint of aluminum shell with different thickness were also studied. The results show that the experimental results of PTFE/Al/W under the impact of drop weight are consistent with the description of the Zerilli–Armstrong constitutive equation, and it was also found that the main reason for the crack generation and propagation in the sample is due to the debonding and separation between the PTFE matrix and W particles. Osborne [10] and Mock [1,11] carried out drop impact test and Taylor impact test for PTFE/Al reactive materials, and the influence of Al particle size on the energy release characteristics of PTFE/Al reactive materials was studied. They found that the PTFE/Al reactive materials with smaller Al particles are more likely to react, because the smaller Al particle size means that the specific surface area of the particles is larger, and the initial energy required to induce the reaction is less.

In practice, the PTFE/Al reactive material needs a lot of external input energy to stimulate its violent chemical reaction, and usually cannot react completely. Ames found that the higher the impact loading speed is, the higher the energy release rate of the material is [12], thus the PTFE/Al reactive material has lower risk than conventional explosives [13]. However, when used in fragment warhead, the PTFE/Al fragment has poor penetration ability and shows little risk to the target if the collision speed is less than 1000 m/s, while the inert fragment still has strong damage power. Thus, PTFE/Al needs to be reformulated to decrease its stimulating energy. The reactive material designed in this paper is based on PTFE as the main component, because PTFE will produce a large amount of oxidizing fluoride when it decomposes. In addition, the reactive metal is also a major component. Through a comprehensive comparative analysis of the physical and chemical properties of common metal powders, it is found that Al metal powder is the most suitable metal powder element in terms of economic cost, safety, and energy release ability. It is worth noting that the non-metallic elements Si and B can exhibit certain metal characteristics, and the corresponding fluorides are gaseous products under normal conditions. Therefore, the appropriate amount of Si and B elements can also be considered when designing the reactive material. In addition, to effectively increase the energy release rate of the PTFE/Al reactive material, it is considered to add proper amount of CuO powder to the PTFE/Al reactive material. Six kinds of reactive materials were designed and tested for their energy release ability under low-speed impact, and the results can provide reference for the formulation design of reactive materials. 

## 2. Formulation Design and Sample Preparation of Reactive Materials

### 2.1. Formulation Design of Reactive Materials

The reactive material designed in this study is mainly composed of PTFE, which will produce a large amount of oxidizing fluoride when it decomposes. To produce a strong oxidation-reduction reaction and release a large amount of heat, it is necessary to select metal powder with strong reduction. Through a comprehensive comparative analysis of the physical and chemical properties of common metal powders, it is found that Al metal powder is the most suitable metal powder elements for the active inner core in terms of economic cost, safety, and energy release capacity. It is worth noting that the non-metallic elements Si and B can exhibit certain metal characteristics, and the corresponding fluorides are gaseous products under normal conditions. Therefore, in this study, the appropriate amount of Si and B elements were also considered when designing the reactive material. In addition, to effectively improve the energy release rate of PTFE/Al reactive materials, nano-sized Al/CuO thermite can be added into the PTFE/Al active materials. Therefore, this study preliminarily determined that the reactive material formulation element components to be used are PTFE, Al, B, Si, and CuO. The relative molecular mass and density of each element component are shown in Table 1.

The chemical reaction equations involved in each element component are as follows:$$4\mathrm{Al}+3C_{2}F_{4}\to 4{\mathrm{AlF}}_{3}+6C$$
$$2\mathrm{Al}+3\mathrm{CuO}\to {\mathrm{Al}}_{2}O_{3}+3\mathrm{Cu}$$
$$4B+3C_{2}F_{4}\to 4{\mathrm{BF}}_{3}+6C$$
$$\mathrm{Si}+C_{2}F_{4}\to {\mathrm{SiF}}_{4}+2C$$

It can be seen from the above reaction equations that PTFE can react with Al, B, and Si, respectively. Therefore, in the formulation design, PTFE/Al, PTFE/B, and PTFE/Si were determined as three independent formulations. The greatest advantage of PTFE/B and PTFE/Si compared to PTFE/Al is the ability to form gaseous products. Based on many literature studies, PTFE/Al is a mature active material formulation for current application and research, but its energy release rate and external functional capacity need to be further improved. Therefore, another design idea of the active material formulation of this study was to use PTFE/Al groups as the basic composition, and then add PTFE/B, PTFE/Si, and Al/CuO groups, respectively. The reactive material still took PTFE as the main component, and the mass ratio between PTFE/Al group and the added group was initially set as 4:1. Thus, the formulation of the six reactive materials designed in this study can be obtained, as shown in Table 2.

### 2.2. Formulation Design of Reactive Materials

The reactive material components designed mainly include PTFE and Al, but, due to the strong hydrophobicity and insolubility of PTFE, the affinity with the Al powder is poor. Therefore, in this study, surface treatment of the metal powder with a silicon coupling agent was selected, and the water repellency or lipophilicity of the Al powder treated by the coupling agent could be improved to some extent. First, take a certain amount of silicon coupling agent (the amount of the silicon coupling agent is usually 0.5%–2.0% of the general filler amount), and dissolve it in absolute ethanol. Then, add Al powder to the solution and let stand for 1 h. Next, heat up and stir until anhydrous ethanol in the solution is completely volatilized. Finally, the remaining material was placed in a vacuum drying oven for drying (about 6 h), and, after the drying was completed, the treated metal powder was taken out.

According to the reactive material formula design, each element component powder was taken according to the mass ratio, and poured into a three-dimensional motion mixer. The mixing time of the three-dimensional motion mixer was about 1 h. Then, the uniformly mixed formulation was sealed and placed in a constant temperature drying oven to prevent oxidation and moisture, and the mixed powder was taken out when the molding was required.

The mass of the desired powder was determined according to the size of the sample, and the powder was poured into a mold for pressing. It is best to press-form at one time to avoid the layered interface. To reduce or avoid the phenomenon of uneven density caused by unidirectional pressing (the density near the punch end is greater than the density away from the punching end), the sleeve direction can be changed during the pressing process to achieve two-way suppression. After the sample is pressed, the punch cannot be loosened; the pressure was maintained for about 1–2 min, and then the punch was released to release the pressure.

Subsequently, the pressed sample was demolded, and, after the sample was demobilized from the mold, it was placed in a normal temperature, normal pressure, and dry environment to release residual stress inside the sample. After the sample was allowed to stand for 24 h, it was sintered. The sintering temperature control curve of the reactive material sample is shown in Figure 1.

## 3. Energy Release Ability Test of Reactive Materials

Based on the previous formulation design, there were six pre-designed reactive material formulations. In this section, we present the screening for the reactive material formulation with the best release ability. The specific test methods mainly included: traditional drop weight test to qualitatively analyze the release energy of reactive materials and the new independent design energy release test device based on the drop weight system to quantitatively analyze the release energy of reactive materials. According to the above two test methods, the reactive material formulation with the best release ability was finally determined.

### 3.1. Qualitative Energy Release Test

The drop weight system is widely used to test various types of energetic materials (such as explosives, polymers/metal powders, etc.). The structural schematic and physical diagram of the drop weight system are shown in Figure 2. The test process can judge whether the reactive material reacts based on fire, sound, smoke, smell, etc. Because the response time scale of the test sample corresponding to the drop weight test is usually small, to be able to analyze the reaction process of the test sample more clearly and carefully, it is usually necessary to record the whole reaction process with the help of high-speed photography in the test process.

To study the energy release ability of the six reactive materials under the condition of drop impact, a batch of samples with the size of ∅ 6 × 3 mm^2^ was prepared. In addition, to ensure scientific and consistent test results, four test samples were prepared for each material formulation. The physical diagram of the sample is shown in Figure 3. Each sample was dimensioned before the drop weight test, and the dimensional parameters and mass of each sample are shown in Table 3.

After analyzing the reaction process captured by high-speed photography, we can get the typical impact reaction process corresponding to each reactive material formulation, as shown in Figure 4.

Through careful analysis of the impact reaction process corresponding to each sintered reactive material formulation in Figure 4, the following conclusions can be obtained:

(1) The reaction processes of #2 (PTFE/B) and #3 (PTFE/Si) were weak; only a very weak flare was generated, which indicates the reaction thresholds of #2 (PTFE/B) and #3 (PTFE/Si) were still high. This impact condition could not to induce a sufficient reaction.

(2) The shock reaction processes of #1 (PTFE/Al), #4 (PTFE/Al/B), #5 (PTFE/Al/Si), and #6 (PTFE/Al/CuO) were intense. In view of the fireball generated by the reaction, the reaction intensity of #5 (PTFE/Al/Si) was the highest. However, the difference in the release energy ability between the formulations could not be specifically determined from the fire alone, because the sizes of the fireballs generated by the impact reactions were similar and could not be judged by sound. Therefore, further quantitative testing was required to screen out the optimal formulation.

### 3.2. Quantitative Energy Release Test

Since the reaction mechanism of the reactive material was different from that of the conventional energetic material such as explosive, the release test method of the active material could not completely copy the test method of the conventional energetic material. To this end, Ames [6] designed a dynamic release test system for reactive materials which can quantitatively characterize the release energy of reactive materials. It can be seen from the analysis that the two test systems (the test system developed by Ames and the traditional drop test system) have their own advantages. The test systems developed by Ames can quantitatively test and characterize the energy release abilities of reactive materials, often requiring higher impact velocities. The traditional drop weight test system can qualitatively compare the energy release capabilities of active materials, and the impact speed is usually low. Therefore, this study used the advantages of the above two test systems to design a new energy release test device suitable for the test requirements in this research, to achieve the measurement and characterization of the release energy of reactive materials at lower impact speed.

The layout of the new energy release test device is shown in Figure 5. The working principle of the device is as follow. When the drop hammer falls freely and strikes the striking column, the striking column will further impact the specimen to be tested placed on the anvil, thus inducing the reaction of the specimen to be tested. The sample to be tested will have gas and product formation while a violent chemical reaction occurs, and the high-pressure gas and reaction product are then released by the piston tube, thereby pushing the piston to move. At this time, the pressure sensor on the inner wall of the chamber can synchronously measure the pressure change in the chamber, and can also infer the magnitude of the force of the piston according to the movement of the piston. When studying the energy release characteristics of the reactive material, the energy release effect of the reactive material can be analyzed not only by the relationship of the pressure measured by the pressure sensor with time, but also the functional force of the active material during the reaction according to the movement of the piston. The results obtained by the two pathways are mutually verified, so that the energy release ability of the active material can be more accurately characterized.

Based on the previous analysis, four active materials were selected as the research objects: #1 (PTFE/Al), #4 (PTFE/Al/B), #5 (PTFE/Al/Si), and #6 (PTFE/Al/CuO). To ensure the scientificity and consistency of the test, three test samples were prepared for each material formulation, and the sample size parameters of each formulation are shown in Table 4.

In the process of energy release ability test, the pressure sensor converted the pressure signal in the quasi-closed container into an electrical signal and recorded it with an oscilloscope. Therefore, the oscilloscope directly recorded the voltage–time curve, and the measured electrical signal needed to be reversed to push out the pressure signal received in the chamber. After data processing, the pressreu–time curve measured for the four different material formulations was obtained, as shown in Figure 6. In addition, the whole process of the test also recorded the piston movement of each group of experiments with high-speed photography. The displacement of the piston at different times was recorded according to the distance calibrated by the background grid. Thus, the piston displacement–time curve corresponding to the four different material formulations could also be obtained, as shown in Figure 7.

By analyzing Figure 6 and Figure 7, the peak pressure ∆*P* of the four material formulations under the impact loading conditions and the time *t* required for the piston to move to the position *x* = 300 mm could be obtained. Thus, the peak pressure ∆*P* and piston motion time *t* were counted, as shown in Table 5.

It can be seen intuitively based on Table 5 that the peak pressure ∆*P* of the chamber corresponding to the PTFE/Al/Si active material formulation was the largest, while the corresponding piston movement time was the smallest (the less time is required for the piston motion within the same distance, the greater is the acceleration obtained by the piston under the action of reaction thrust). Therefore, the above two phenomena can well prove that the release capacity of the PTFE/Al/Si formulation was the strongest. The release ability of the four formulations was obtained as follows: PTFE/Al/Si > PTFE/Al/ CuO > PTFE/Al/B > PTFE/Al.

In summary, based on the conclusions obtained from qualitative analysis and quantitative analysis, it can be finally determined that the optimal reactive material formulation designed in this study is PTFE/Al/Si formulation.

## 4. Conclusions

Under the action of high-speed dynamic loading, the reactive materials will undergo intense and rapid combustion or a detonation-like reaction accompanied by the release of a large amount of chemical energy. In this study, six different reactive material formulations (PTFE/Al, PTFE/B, PTFE/Si, PTFE/Al/B, PTFE/Al/Si, PTFE/Al/CuO) were designed. To select the best reactive material formulation, the qualitative and quantitative tests were carried out successively, obtaining the following conclusions.

(1)The drop weight test was carried out on the six reactive material formulations, and the energy release ability of the six reactive material formulations was qualitatively compared. The results show that the reaction processes of #2 (PTFE/B) and #3 (PTFE/Si) were weak with only a very weak flare generated, while the impact reaction processes corresponding to #1 (PTFE/Al), #4 (PTFE/Al/B), #5 (PTFE/Al/Si), and #6 (PTFE/Al/CuO) were intense. However, the difference in the energy release ability among the formulations could not be specifically determined from the fire alone, thus it was necessary to select the best formulation through further quantitative comparison.(2)Based on the conventional drop weight system, a new type of energy release testing device was self-designed, which can quantitatively compare and analyze the energy release ability of reactive materials. The results show that the peak pressure of the chamber corresponding to the PTFE/Al/Si formulation was the largest, and the corresponding piston movement time was the smallest. Therefore, the above two phenomena can well prove that the energy release ability of the PTFE/Al/Si formulation was the strongest, and the energy release ability of the four groups of formulations could be obtained as following order: PTFE/Al/Si > PTFE/Al/ CuO > PTFE/Al/B > PTFE/Al.

Based on qualitative and quantitative analyses, it could be finally determined that the optimal reactive material formulation designed was PTFE/Al/Si formulation. The formulation of the material exhibits excellent properties in terms of energy release ability and may be a new choice for reactive fragment for its lower stimulating energy.

## Figures and Tables

**Figure 1 materials-12-03940-f001:**
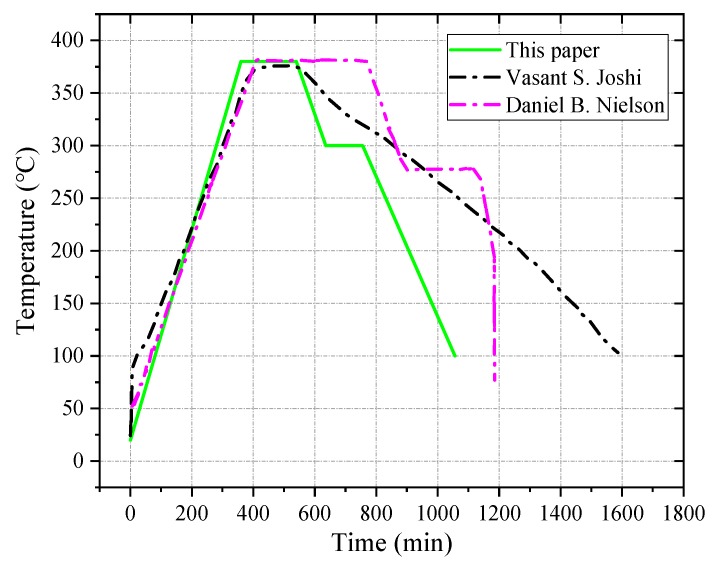
Sintering temperature control curve of the reactive material sample.

**Figure 2 materials-12-03940-f002:**
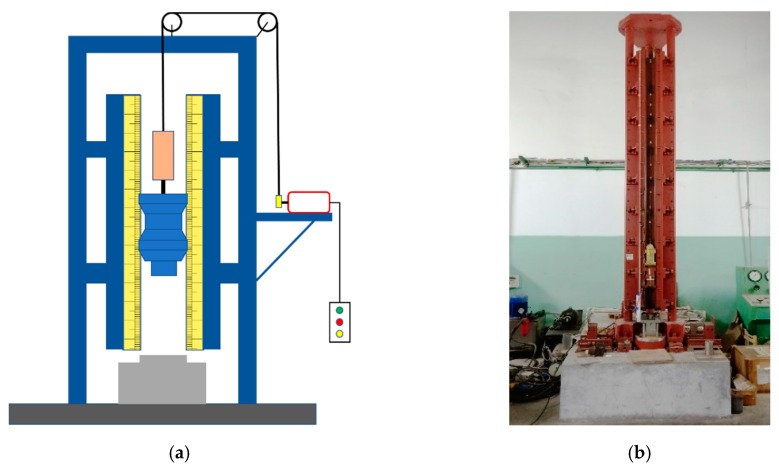
Structural diagram and physical image of the drop weight system: (**a**) structural diagram; and (**b**) physical image.

**Figure 3 materials-12-03940-f003:**
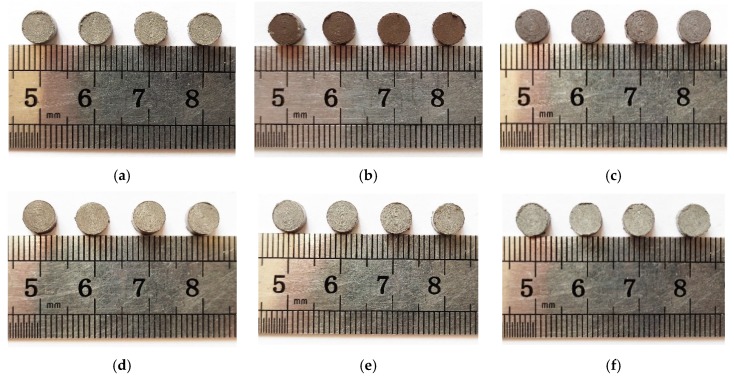
Physical drawing of the reactive material sample (∅ 6 × 3 mm^2^): (**a**) PTFE/Al; (**b**) PTFE/B; (**c**) PTFE/Si; (**d**) PTFE/Al/B; (**e**) PTFE/Al/Si; and (**f**) PTFE/Al/CuO.

**Figure 4 materials-12-03940-f004:**
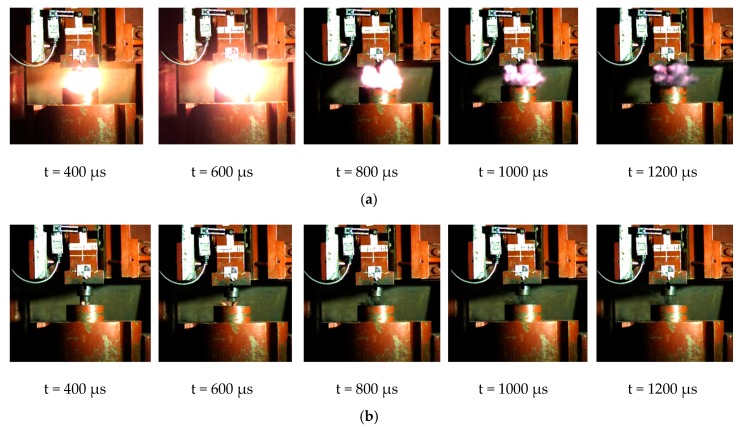
Typical impact reaction process corresponding to each reactive material formulation: (**a**) #1-3 (PTFE/Al); (**b**) #2-1 (PTFE/B); (**c**) #3-3 (PTFE/Si); (**d**) #4-1 (PTFE/Al/B); (**e**) #5-1 (PTFE/Al/Si); and (**f**) #6-1 (PTFE/Al/CuO).

**Figure 5 materials-12-03940-f005:**
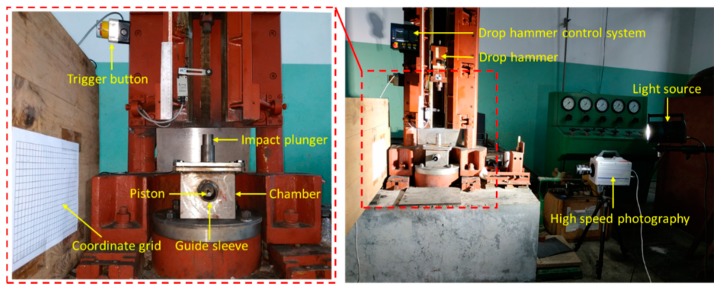
Layout of the new energy release test device.

**Figure 6 materials-12-03940-f006:**
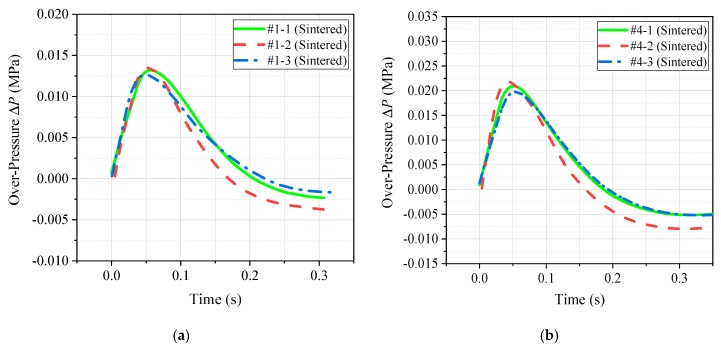
The pressreu–time curve measured by four different material formulations: (**a**) PTFE/Al; (**b**) PTFE/Al/B; (**c**) PTFE/Al/Si; and (**d**) PTFE/Al/CuO.

**Figure 7 materials-12-03940-f007:**
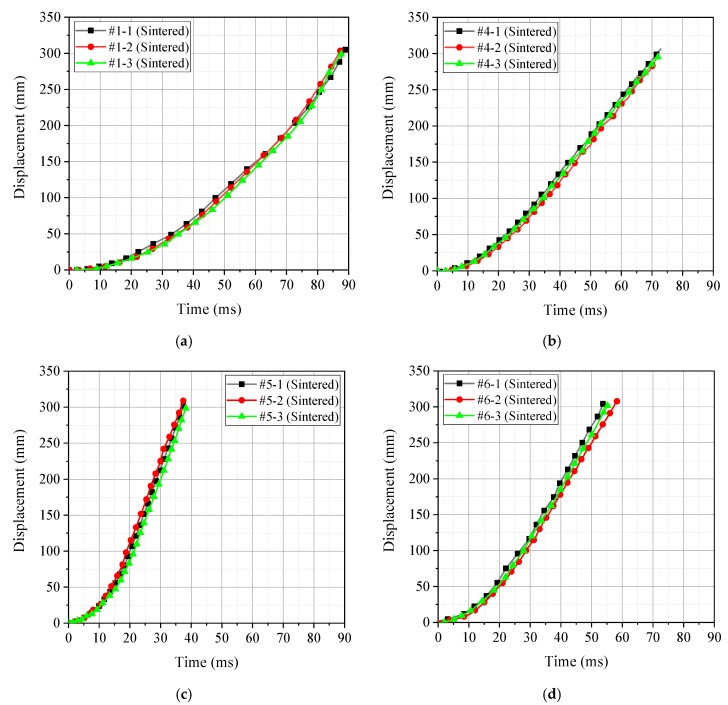
The piston displacement–time curve corresponding to the four different material formulations: (**a**) PTFE/Al; (**b**) PTFE/Al/B; (**c**) PTFE/Al/Si; and (**d**) PTFE/Al/CuO.

**Table 1 materials-12-03940-t001:** Relative molecular mass and density of each element component.

Element Component	Relative Molecular Mass	Density (g/cm^3^)
PTFE	100	2.1
Al	27	2.7
B	11	2.46
Si	28	2.33
CuO	80	6.4

**Table 2 materials-12-03940-t002:** Formulation of reactive materials.

Number	Formulation	Component Percentage (%)	Remark
#1	PTFE/Al	PTFE:Al = 73.5:26.5	-
#2	PTFE/B	PTFE:B = 87.2:12.8	-
#3	PTFE/Si	PTFE:Si = 78.1:21.9	-
#4	PTFE/Al/B	PTFE:Al:B = 76.2:21.2:2.6	(PTFE/Al):(PTFE/B) = 4:1
#5	PTFE/Al/Si	PTFE:Al:Si = 74.4:21.2:4.4	(PTFE/Al):(PTFE/Si) = 4:1
#6	PTFE/Al/CuO	PTFE:Al:CuO = 58.8:24.9:16.3	(PTFE/Al):(Al/CuO) = 4:1

**Table 3 materials-12-03940-t003:** Dimensional parameters and mass of each sample (qualitative test).

Number	Formulation	Mass (g)	Diameter (mm)	Thickness (mm)
#1-1	PTFE/Al	0.184	6.04	3.10
#1-2		0.181		3.04
#1-3		0.182		3.08
#1-4		0.189		3.10
#2-1	PTFE/B	0.175		2.90
#2-2		0.170		2.90
#2-3		0.180		3.10
#2-4		0.182		3.12
#3-1	PTFE/Si	0.183		3.10
#3-2		0.187		3.14
#3-3		0.187		3.14
#3-4		0.174		3.00
#4-1	PTFE/Al/B	0.194		3.14
#4-2		0.191		3.14
#4-3		0.190		3.08
#4-4		0.192		3.10
#5-1	PTFE/Al/Si	0.185		2.96
#5-2		0.195		3.10
#5-3		0.197		3.16
#5-4		0.198		3.22
#6-1	PTFE/Al/CuO	0.204		3.08
#6-2		0.210		3.12
#6-3		0.208		3.02
#6-4		0.209		3.10

**Table 4 materials-12-03940-t004:** Dimensional parameters and mass of each sample (quantitative test).

Number	Formulation	Mass (g)	Diameter (mm)	Thickness (mm)
#1-1	PTFE/Al	0.175	6.04	2.94
#1-2		0.175		2.98
#1-3		0.174		3.02
#4-1	PTFE/Al/B	0.196		3.12
#4-2		0.196		3.18
#4-3		0.197		3.14
#5-1	PTFE/Al/Si	0.191		3.04
#5-2		0.191		2.98
#5-3		0.191		3.02
#6-1	PTFE/Al/CuO	0.206		2.96
#6-2		0.206		2.98
#6-3		0.205		3.03

**Table 5 materials-12-03940-t005:** Peak pressure and piston motion time.

Formulation	Peak Pressure ∆*P* (MPa)	Piston Motion Time *t* (ms)
PTFE/Al	0.013	88
PTFE/Al/B	0.021	72
PTFE/Al/Si	0.051	37
PTFE/Al/CuO	0.039	55

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
