# Peer review of "Research on the Energy Release Characteristics of Six Kinds of Reactive Materials"

_materials, 2019, doi:10.3390/ma12233940_

Round 1

Reviewer 1 Report

In the manuscript entitled "Research on the Energy Release Characteristics of Six Kinds of Reactive Materials" the Authors investigated six different PTFE/metal and non metal mixtures.

Test methodology and sample preparation was adequately described and also the results discussion is supported by data obtained during investigation. There is one thing I find missing, the adiabatic combustion temperature. If the Authors could measure it, it would greatly improve the overall merit of the manuscript, as adiabatic temperature is a very important parameter when it comes to discuss thermite mixtures.

There is just one issue that I find curious. Is the peak pressure data (Fig. 8, Table 5) correct? The values seem really low, especially for the most potent compositions like PTFE/Al/Si, where it looks like an order of magnitude too low, especially for nanothermite composition.

Author Response

Point 1: There is one thing I find missing, the adiabatic combustion temperature. If the Authors could measure it, it would greatly improve the overall merit of the manuscript, as adiabatic temperature is a very important parameter when it comes to discuss thermite mixtures.

Response 1: It’s right that the adiabatic temperature is important, and we tried to find one appropriate pyrometer within the scope of our city, but unfortunately we found none. Furthermore, the brightness theoretically increase in proportion to the adiabatic temperature, so we think that it is adequate to judge qualitatively with the brightness.

Point 2: There is just one issue that I find curious. Is the peak pressure data (Fig. 8, Table 5) correct? The values seem really low, especially for the most potent compositions like PTFE/Al/Si, where it looks like an order of magnitude too low, especially for nanothermite composition.

Response 2: All the sample used in drop hammer experiments are small and weight no more than 0.3 gram, comparing with the sample weight about 10 gram used in experiments such as Ames(reference 6). At the same time, the impact of drop hammer can not stimulate the sample completely, so it is right that the peak pressure data is low about an order of magnitude with that of Ames, but the difference among six materials can be distinguished adequately.

Reviewer 2 Report

The abstract fails to clearly state the need for this study.  A publication should address a scientific issue that is important, that has not been treated satisfactorily so far, and that cannot be treated using existing approaches.  If the objective is to evaluate various materials its value as a scientific publication is greatly diminished.

Some sentences are very long and contorted, making them hard to follow.  For example: “Since the discovery of PTFE 35 and reactive materials powder in high-speed impact load can react for more than 40 years.  Short declarative sentences are preferred.

The acronym PTFE is used many times.  It should be defined at the start.

While 13 references are cited, the introduction does not really present the state of the art in a   particular area.  This means that it also does not outline a need for further study and does not clearly show how the present work would add to what is already known.

What is to be seen in Fig. 1?  It just looks like small amounts of powder.

Reading through the body of the paper and the conclusion, it appears that this is a ice engineering study.  However the issues raised above still remain: what is the scientific question addressed here?  What is new here?  It seems like a typical approach where one makes some samples, performs some tests and selects the most promising material.

Author Response

Point 1: The abstract fails to clearly state the need for this study.  A publication should address a scientific issue that is important, that has not been treated satisfactorily so far, and that cannot be treated using existing approaches.  If the objective is to evaluate various materials its value as a scientific publication is greatly diminished.

Response 1: The abstract has been rewritten to clarify the need for this study.

Point 2: Some sentences are very long and contorted, making them hard to follow.  For example: “Since the discovery of PTFE 35 and reactive materials powder in high-speed impact load can react for more than 40 years.  Short declarative sentences are preferred.

Response 2: Some long sentences in the paper have been changed to the short one.

Point 3: The acronym PTFE is used many times.  It should be defined at the start.

Response 3: Has been changed according to the comment.

Point 4: While 13 references are cited, the introduction does not really present the state of the art in a   particular area.  This means that it also does not outline a need for further study and does not clearly show how the present work would add to what is already known.

Response 4: The instruction has been rewritten accordingly.

Point 5: What is to be seen in Fig. 1?  It just looks like small amounts of powder.

Response 5: The fig.1 has been deleted

Point 6: Reading through the body of the paper and the conclusion, it appears that this is a ice engineering study.  However the issues raised above still remain: what is the scientific question addressed here?  What is new here?  It seems like a typical approach where one makes some samples, performs some tests and selects the most promising material.

Response 6: This point 6 is same with point 1 and point 4, and the paper has been rewritten according to the two points.